# Self-Perceived Health, Mood, and Substance Use Among Adolescents: An Analysis to Enhance Family, Community, and Mental Health Care

**DOI:** 10.3390/healthcare12222304

**Published:** 2024-11-18

**Authors:** José Antonio Zafra-Agea, Cristina García-Salido, Estel·la Ramírez-Baraldes, Mireia Vilafranca-Cartagena, Ester Colillas-Malet, Anna Portabella-Serra, Daniel García-Gutiérrez

**Affiliations:** 1Department of Nursing, Faculty of Health Sciences at Manresa, University of Vic–Central University of Catalonia (UVic-UCC), Av. Universitària, 4-6, 08242 Manresa, Spain; 2Department of Nursing, Faculty of Nursing, Physiotherapy, and Podiatry, University of Seville, 41004 Seville, Spain; 3CAP Verdaguer, Primary Care and Community Management Baix Llobregat, Catalan Institute of Health, University Institute for Primary Care Research (IDIAP Jordi Gol), 08007 Barcelona, Spain; 4Research Group on Simulation and Transformative Innovation (GRITS), Institute of Research and Innovation in Life and Health Sciences of Central Catalonia (IRIS-CC), 08242 Manresa, Spain; 5Intensive Care Unit, Althaia University Health Network, 08242 Manresa, Spain; 6Research Group on Epidemiology and Public Health in the Context of Digital Health (Epi4Health), Institute of Research and Innovation in Life and Health Sciences of Central Catalonia (IRIS-CC), 08242 Manresa, Spain; 7Internal Medicine, Althaia University Health Network, 08243 Manresa, Spain

**Keywords:** adolescents, substance use, emotional well-being, mental health, community care, public health policies, prevention

## Abstract

Background: Adolescence is a critical period for developing self-perception, emotional well-being, and health behaviors. Mental health disorders represent a substantial burden for adolescents worldwide. This study examines self-perceived health, mood, and substance use among adolescents, identifying associated risk factors. Method: A cross-sectional study was conducted with 121 adolescents aged from 14 to 18 from a secondary school in Baix Llobregat, Catalonia. Data were collected through questionnaires, and descriptive and comparative analyses were performed. Results: Poor self-perceived health and negative mood were associated with higher alcohol and tobacco use. Girls exhibited better emotional regulation than boys. Conclusions: Poor health perception and negative mood are linked to increased substance use. Early intervention should focus on emotional well-being and prevention, involving both families and schools.

## 1. Introduction

Adolescence, a transitional phase between childhood and adulthood, is marked by significant physical, emotional, social, and cognitive changes [1,2]. These transformations are not only related to biological development but are also deeply influenced by environmental factors, including family dynamics, peer interactions, and social expectations. During this stage, adolescents begin to forge their identity, which significantly influences their behaviors, attitudes, and self-perception [3].

Several critical factors influence the formation of healthy behaviors during adolescence, such as perceived health, emotional well-being, and body image [3]. These elements are closely interrelated, with each playing a crucial role in the adolescent’s overall development. Self-perceived health is especially influential; how adolescents view their own health can affect their likelihood of engaging in risky behaviors, such as substance use, and impacts their emotional resilience and mental well-being [4].

Body image, often shaped by social pressures and comparisons with peers, can have a profound influence on self-esteem and mental health, particularly during adolescence, where physical appearance becomes central to social interactions [5]. A negative body image is closely associated with mental health disorders such as anxiety, depression, and eating disorders, which are common during this stage of development [6].

Substance use is one of the most concerning risk behaviors during adolescence, as it can have lasting impacts on both physical and mental health. Early initiation of alcohol, tobacco, and other drugs is consistently associated with higher levels of addiction in adulthood. Recent data show that substance-use prevalence among secondary-school students remains alarmingly high, with 73.6% for alcohol, 27.7% for tobacco, and 21.8% for cannabis [7]. The early age of initiation, particularly for alcohol and tobacco, exacerbates the risk of addiction and other long-term health problems. Moreover, gender differences in substance use highlight the need for targeted prevention strategies: while women tend to have higher rates of alcohol, tobacco, and sedative consumption, men report a higher prevalence of daily alcohol consumption [8,9].

Mental health disorders represent a substantial burden for adolescents worldwide. Globally, one in seven adolescents aged from 10 to 19 suffers from some form of mental disorder, with depression and anxiety being the most common. These disorders are not only debilitating during adolescence but can also have long-term effects, persisting into adulthood and affecting individuals’ ability to lead productive and fulfilling lives [10]. The impact of untreated mental health issues is wide-ranging, contributing to elevated rates of disability and even mortality, with suicide being the fourth leading cause of death among young people aged from 15 to 29 [7]. The long-term consequences of neglecting adolescent mental health underscore the importance of early intervention and comprehensive mental health services [11,12].

Socioeconomic disparities have a profound impact on adolescent health, influencing not only perceived health but also access to health services and resources. Data show that adolescents from lower socioeconomic backgrounds are more likely to report poor or very poor health. In 2023, 18.6% of adolescents from lower socioeconomic classes reported poor health, compared to only 10.7% of those from higher socioeconomic classes. This disparity is reflected in other health outcomes, including higher rates of mental health issues and substance use in disadvantaged communities. Addressing these inequalities is crucial for developing effective public health interventions [13].

The school environment plays a fundamental role in adolescent development, not only as an educational space but also as a key setting for promoting health and well-being. Schools not only facilitate academic learning but can also be health-promoting environments, helping to develop emotional competencies, social skills, and healthy behaviors. The group of 4ths ESO adolescents can be considered “sentinel agents” when studying health-related behaviors, as many habits acquired during this stage may solidify and persist throughout life [14].

In this context, periodic surveys on health-related habits are essential tools for monitoring adolescent behaviors. In some municipalities, such as those supported by the Public Health Service of the Diputació de Barcelona, surveys have been conducted with 4th ESO students to assess the health situation and design specific interventions. These surveys, which explore areas such as health perception, eating habits, sexuality, mood, and interpersonal relationships, have served as a basis for guiding and evaluating local health policies [15].

Local governments and community organizations are key actors in addressing adolescent health challenges. Municipal policies aimed at regulating access to addictive substances, promoting mental health services, and creating safe and supportive environments are essential for preventing substance use and promoting overall well-being. These policies should be based on local data and tailored to the specific needs of the community, ensuring that interventions are culturally appropriate and accessible to all adolescents, regardless of socioeconomic status.

In addition to direct health interventions, municipal policies can support the creation of “health-promoting environments”, including safe recreational spaces, access to mental health resources, and community programs that engage adolescents and their families. Collaboration between schools, health services, and community organizations is vital to developing integrated adolescent health approaches that address both individual and systemic factors [16].

The present study aims to describe self-perceived health, mood, body image, and substance use among 4th-year secondary school students in a school and community not previously studied. By examining these factors alongside the influence of social and municipal policies, the study seeks to provide a comprehensive overview of adolescent health and well-being. Additionally, the study highlights the importance of preventive programs that engage both families and communities, fostering environments that promote resilience and reduce risky behaviors, as such programs have shown great efficiency in other student populations.

Through an integrative approach, this initial study will provide valuable data that can inform the development of targeted local interventions. By identifying key risk factors and protective elements, the findings will contribute to the design of more effective public health strategies aimed at improving adolescent health outcomes. Furthermore, the study emphasizes the need for a multidimensional approach to health promotion, involving schools, families, communities, and public administrations in a coordinated effort to improve young people’s quality of life.

## 2. Methodology

### 2.1. Study Design

This study employed a cross-sectional descriptive design with a quantitative approach, conducted in a municipality in the province of Barcelona, which includes centers of public and private-concerted ownership guaranteeing a representative sample of the 563 ESO students, during the 2023–2024 academic year. The aim of the study was to analyze the prevalence of habits and behaviors related to self-perceived health, mood, body image, and substance use among adolescents enrolled in the 4th year of secondary education, and it represents the total population of the only school in a municipality of 13,000 inhabitants.

### 2.2. Study Population

The study population included 121 students in the 4th year of ESO at a selected school. Participants completed an online validated questionnaire, which was self-administered in educational centers individually and anonymously during class hours, after obtaining informed consent from their legal guardians. Participation in the study was voluntary, and data were collected confidentially and used only in aggregate form.

Response rate: Of the students invited, 86.3% completed the survey. This percentage represents the effective response rate for the sample.

Differences between respondents and non-respondents: The initial analysis did not include an evaluation of the differences between students who responded and those who did not. Due to the anonymous and confidential nature of the survey, as well as the data-collection process, a detailed analysis to identify significant differences between these groups was not conducted.

### 2.3. Ethical Procedures

This study was conducted in accordance with the ethical guidelines outlined in the Declaration of Helsinki. Informed consent was obtained from the legal guardians to ensure the voluntary, anonymous, confidential participation of the students. The study was approved by the educational institution’s administration, passed the institutional ethics committee review, and received approval from the local public health service, which was the driving force behind the study, with approval number SP:23/002 (19 September 2023). Each participant provided written informed consent.

### 2.4. Data-Collection Instruments

An online questionnaire consisting of 76 questions, provided by the Barcelona Provincial Council, was utilized to assess health habits among 4th-year secondary school students in the province of Barcelona [14]. The questionnaire addressed various health-related domains, including self-perceived health; BMI; weight perception; mood; and substance use, encompassing tobacco, alcohol, cannabis, cocaine, and other drugs. The aim of the study was to analyze the prevalence of habits and behaviors related to self-perceived health, mood, body image, and substance use among adolescents in 4th-year secondary school to identify risk factors, compare behaviors, monitor these behaviors, and evaluate their impact.

### 2.5. Variable Recategorization

The variables used for this study were recategorized by grouping responses to ensure that each category contained at least 10% of the cases.

### 2.6. Data-Collection Procedure

The survey was distributed electronically to students during class hours. Data collection took place between November 2023 and February 2024.

### 2.7. Statistical Analysis

Descriptive statistics were calculated, including frequencies, percentages, means, and standard deviations (SDs). Pearson’s chi-square test was used to compare groups and assess associations between demographic characteristics, behaviors related to self-perceived health, mood, and substance use among adolescents. A *p*-value of <0.05 was considered statistically significant. All analyses were performed using R statistical software v. 4.4.1 (R Project for Statistical Computing).

## 3. Results

### Sample Description

The sample comprised 121 students from the fourth year of secondary education (4th ESO) at a public high school in a town in the province of Barcelona, aged between 14 and 18 years. The average age was 15.2 years, with a gender distribution of 57.3% female and 42.7% male. Most of the students were native (71.8%), with 3.2% being first-generation immigrants, 16.9% second-generation immigrants, and 8.1% who did not specify their background.

Regarding the family structure, 71% had a biparental family, 24.2% had a single-parent family, and 3.2% had a restructured family. Socioeconomic status was classified as high for 63.7%, medium for 29.8%, and low for 6.5%.

The educational levels of the parents showed that most had secondary education (39.1%) or university degrees (28.2%). A small percentage had no formal education (5.6%) or primary education (8.5%), and 18.6% did not specify or were unsure.

Table 1 shows the percentage distribution of self-perceived health by gender. No significant differences were found in health perception between men and women in the categories of “very good” (*p* = 0.620), “good” (*p* = 0.879), and “bad or very bad” (*p* = 0.723), while there was a trend towards a higher perception of “regular” health in women (15.9%) compared to men (7.8%), with a *p*-value approaching the significance threshold (*p* = 0.095).

In Table 2, the frequencies and percentages of various emotional states reported by students are presented. A total of 53.3% of adolescents indicated feeling “always” or “often” too tired to engage in activities, suggesting a high prevalence of fatigue within this population. Additionally, 31.7% of participants reported frequently having trouble sleeping, reflecting the existence of sleep-related difficulties.

Regarding emotional distress, 48.3% of students stated they “sometimes” or more frequently felt displaced, sad, or depressed. Similarly, 47.5% expressed feelings of hopelessness about the future, which could be indicative of concerns related to their psychological well-being and expectations.

Nervousness or tension was reported by 40.9% of respondents, who felt this way “always” or “often”. Additionally, 47.5% of students reported frequent boredom, suggesting a potential disconnection or lack of engagement with their environment.

Lastly, although less prevalent, 50% of adolescents reported feeling angry or violent “sometimes” or more frequently, highlighting potential challenges in emotional regulation and anger management within this group.

Table 3 presents the analysis of responses to the questions “Have you harmed yourself intentionally?” and “Have you thought about wanting to die?” by gender.

Regarding the question “*Have you harmed yourself intentionally?*” the distribution of responses indicates no significant gender differences in the “never” category (*p*-value = 0.410). Boys reported 33 instances (70.6%), and girls reported 20 instances (47.8%), suggesting that the frequency of not harming oneself intentionally does not significantly vary between boys and girls. For the categories “occasionally” and “frequently”, no significant differences were found (*p*-value = 0.496 and *p*-value = 1.000, respectively). This indicates that the occurrence of occasional and frequent self-harm is similar across genders.

In the categories of “almost every day”, “don’t know”, and “prefer not to say”, the results also show no significant differences (*p*-value = 0.157, *p*-value = 0.479, and *p*-value = 0.527, respectively). The low frequency of self-harm almost every day and the tendency to avoid answering do not vary significantly by gender. In summary, for most categories, gender does not have a significant impact on the distribution of responses regarding intentional self-harm.

For the question “*Have you thought about wanting to die*?” no significant differences were found in the categories of “never”, “occasionally”, and “frequently” (*p*-value = 0.321, *p*-value = 0.454, and *p*-value = 1.000, respectively). The frequency of never having thought about wanting to die, as well as occasional and frequent thoughts about dying, does not significantly differ between boys and girls. Similarly, in the “almost every day” category, no significant difference was observed (*p*-value = 0.479), indicating that the frequency of thoughts about dying almost every day is similar across genders.

A significant difference was found in the “don’t know” category (*p*-value = 0.005), with girls reporting eight instances (5.8%), while boys reported none. This suggests that a higher proportion of girls are uncertain about their thoughts of dying compared to boys. In the “prefer not to say” category, no significant difference was found (*p*-value = 0.740), indicating that the tendency to avoid answering is similar between boys and girls. In summary, gender does not significantly affect the distribution of responses in most categories, except for uncertainty about suicidal thoughts, where a higher proportion of girls exhibit uncertainty compared to boys.

Table 4 presents the results for the question “*How do you feel about your own body?*” and analyzes the results by gender. The *p*-values indicate significant differences in body satisfaction between boys and girls. In the “very satisfied” category, boys report 70.6%, compared to 47.8% for girls (*p*-value < 0.001). This indicates that a higher proportion of boys are very satisfied with their bodies compared to girls. Similarly, in the “relatively dissatisfied” and “very dissatisfied” categories, girls show higher percentages (10.1% and 8.7%, respectively) compared to boys (5.9% and 2%, respectively), with *p*-values < 0.001. This suggests that girls tend to express greater dissatisfaction with their own bodies compared to boys.

Regarding opinions about weight, the results for the question “*How do you feel about your weight?*” reveal significant differences between boys and girls. In the “appropriate” category, a higher percentage of boys (66.7%) consider their weight appropriate compared to 56.5% of girls (*p*-value = 0.005). On the other hand, in the “too high” category, girls (8.7%) report a higher perception of their weight being high compared to boys (0%) (*p*-value = 0.018). These differences suggest that girls tend to have a more critical perception of their weight compared to boys, reflecting a greater concern among girls about their weight.

BMI (Body Mass Index) examines the perception of BMI based on the question “*How would you describe your BMI?*” The *p*-values reveal significant differences in BMI perception between boys and girls. Boys report a higher percentage in the “normal weight” category (68.6%) compared to girls (60.9%) (*p*-value = 0.025). Additionally, a higher percentage of girls (7.2%) report obesity compared to boys (0%) (*p*-value = 0.032). These differences reflect the fact that girls tend to perceive their BMI as less healthy compared to boys, with greater concern about weight and obesity.

In the analysis of the initiation of substance use, it is observed that many participants began smoking at ages 13 or 14, with 29.3% reporting initiation at each of these ages, indicating that these are the most common ages for starting tobacco use in this sample. A significant 14.6% began smoking at ages 12 and 15; however, this is less prevalent compared to ages 13 and 14. Only 2.4% began smoking at age 16 or older, suggesting that most individuals start much earlier than this age. Additionally, 4.9% of participants either did not know or did not respond to the question, which may be due to difficulties in recalling or reluctance to disclose this information.

Table 5 analyzes the prevalence of smoking initiation and current smoking status by gender. The analysis reveals a significant difference in the prevalence of having ever smoked between boys and girls. Specifically, 42.15% of girls reported having smoked, compared to only 19.83% of boys. In contrast, a larger percentage of boys (80.4%) have never smoked, compared to 58.33% of girls. The *p*-value for this comparison is less than 0.001, indicating a statistically significant difference between genders.

Regarding current smoking status, among those who have smoked before, there is no significant gender difference in smoking status. Among boys who smoke, 41.67% are currently smoking, while 58.33% are not. Among girls who smoke, 37.25% are currently smoking, and 62.75% are not. The *p*-value for this comparison is approximately 0.539, which is above the typical significance threshold of 0.05. This suggests that there are no statistically significant differences in current smoking status between boys and girls who have previously smoked.

The distribution of smoking frequency among participants who currently smoke. Among those who smoke, 31.3% report smoking daily, indicating a significant proportion with a habitual daily smoking pattern. Additionally, 18.8% of participants smoke more than twice a week, reflecting a relatively frequent but less regular consumption. A smaller percentage of participants, 12.5%, smoke 1–2 times a week, showing an intermediate level of smoking frequency. Notably, the largest proportion of participants, 37.5%, smoke less than once a week, suggesting that although these individuals are categorized as current smokers, their consumption is relatively infrequent. This distribution illustrates the variability in smoking habits among current smokers, with a substantial number engaging in daily smoking, while others smoke much less frequently.

Table 6 analyzes Prevalence of alcohol consumption by gender, the analysis the *p*-value of 0.003 for both categories (“yes” and “no”) indicates a statistically significant difference in the prevalence of alcohol consumption between boys and girls. This suggests that girls have a significantly higher rate of alcohol consumption compared to boys, revealing significant differences in alcohol-consumption patterns based on previous alcohol use. No individuals who have ever consumed alcohol do so during class days, whereas 98.9% of those who have never consumed alcohol do. The *p*-value for this comparison is less than 0.001, indicating a statistically significant difference. For weekends, 24.7% of those who have consumed alcohol drink during this time, compared to 74.2% of those who have never consumed alcohol. This difference, with a *p*-value of 0.001, also signifies statistical significance. At family parties, 69.7% of those who have ever consumed alcohol drink, whereas only 29.2% of those who have do not. The *p*-value is less than 0.001, showing a significant difference. Lastly, 51.7% of those who have consumed alcohol do so at discos or bars, compared to 47.2% of those who have never consumed alcohol, with a *p*-value of 0.001, indicating a statistically significant difference.

We asked the question “Have you consumed 4 or more drinks on one occasion?” Among the respondents, 37.1% reported having consumed four or more drinks on one occasion, while 48.3% reported not doing so. The remaining 12.4% could not recall. The *p*-value calculation for this question would require further breakdown by gender or other variables to determine statistical significance.

We asked the question “Have you been drunk?” Regarding the frequency of drunkenness, 31.5% reported being drunk more than twice, 11.2% reported being drunk twice, 13.5% once, and 43.8% never. Like the previous question, a detailed statistical analysis by gender or other factors is needed to assess the significance of these patterns.

The analysis of alcohol-consumption categories reveals no statistically significant differences between boys and girls in their alcohol-related experiences. In the category of having been drunk four or more times, the percentages for boys (34.5%) and girls (35.1%) are nearly identical, with a *p*-value of 0.939 indicating no significant difference. Similarly, the *p*-value of 0.637 for having been drunk in general suggests that intoxication rates do not differ significantly between genders. Regarding the consumption of four or more drinks on one occasion, the *p*-value of 0.216 indicates that the difference between 0% of boys and 5.3% of girls is not significant. The proportion of individuals who have never been drunk is comparable between boys (31%) and girls (33.3%), with a *p*-value of 0.751, indicating no significant difference in this category. Finally, the *p*-value of 0.271 for non-responses shows that the lack of response is similar between genders. These results suggest that patterns of alcohol consumption are quite similar between boys and girls in this sample.

Table 7 reveals significant differences in the use of certain addictive substances between boys and girls. Specifically, 23.2% of girls reported using hashish or marijuana, compared to 13.7% of boys, with a *p*-value of 0.029 indicating a statistically significant difference. This suggests a higher prevalence of hashish or marijuana use among girls. For tranquilizers, 11.6% of girls have used them, compared to 5.9% of boys, with a *p*-value of 0.083. Although this *p*-value is near the threshold of significance, it does not reach conventional statistical significance, suggesting a possible trend towards higher usage among girls but not a definitive difference.

Regarding the use of cocaine, ecstasy, and inhalants, the data show that consumption is very low in both genders. Specifically, none of the boys has used cocaine or ecstasy, and only 2.9% of girls have used inhalations, with *p*-values of 0.222 and 0.135, respectively. These *p*-values indicate that the differences observed are not statistically significant. Lastly, a higher percentage of boys (78.4%) report not using any substances compared to girls (66.7%), with a *p*-value of 0.073. This *p*-value suggests that while the difference is not statistically significant, there is a trend indicating that boys are more likely to abstain from substance use. The perception of the danger of substances shows that 14.5% of respondents consider hashish or marijuana to be dangerous, while 15.3% perceive tranquilizers as dangerous substances. In contrast, the remaining substances are considered highly dangerous by an average of 72% of respondents.

## 4. Discussion

Understanding the relationship between self-perceived health, mood, body image, and substance use among 4th ESO students within a school and community context is essential to emphasize the importance of preventive programs that involve both families and communities. These programs help create environments that foster resilience and reduce risk behaviors.

In relation to self-perceived health, our adolescents (aged 14–18) report a high prevalence of fatigue, often manifested through sleep problems. It is crucial to differentiate whether the adolescent experiences excessive sleepiness or fatigue to provide appropriate treatment [17,18,19] and implement effective prevention strategies, as fatigue is a common symptom, reported by 7–25% in various studies [20]. The primary cause of daytime sleepiness is insufficient or poor-quality sleep [21]. Fatigue, defined as subjective tiredness, lack of energy, and excessive exhaustion, is shown in several studies, with a variable prevalence between 0.7 and 7.4% [22].

Regarding emotional distress, 48.3% of the adolescents reported feelings of depression, sadness, and nervousness; 47.5% expressed concerns about their psychological well-being and future expectations, indicating a possible disconnection or lack of engagement with their surroundings. A study conducted in our country involving ESO students from the autonomous communities of the Principality of Asturias and La Rioja (n = 508) revealed that a significant number of adolescents self-reported emotional symptoms, with distinct differences based on gender and age [23]. The findings indicated that between 8.7 and 22.6% of students reported emotional symptoms, while 2.4–14.6% exhibited behavioral symptoms. Based on these results, the study recommended that addressing mental health should be a priority in government health policies. Children and adolescents exhibiting significant symptoms of depression and anxiety also reported greater impairment in daily activities due to somatic complaints [24].

In the section on self-harm and suicidal thoughts, a low frequency of self-harm was observed among the surveyed students, with no significant gender differences. However, it was notable that females expressed greater uncertainty about their thoughts related to death compared to males. A study involving Spanish adolescents (n = 1664) concluded that suicidal ideation and self-harm are associated with lower subjective emotional well-being and greater emotional and behavioral problems [25].

Body image refers to the mental representation one has of their body. Adolescence is a critical period where various bio-psycho-social changes occur, influencing the formation of body image both cognitively and subjectively, as well as affecting adaptation to social environments and group identity [26]. In this study, 70.6% of boys, compared to 47.8% of girls (*p*-value < 0.001), reported being satisfied with their body, indicating that boys are generally more satisfied with their body image. Girls, on the other hand, are more concerned about their weight and often perceive their BMI as less healthy. A contrasting study involving 189 adolescents (ages 12–16) found that 87.65% were dissatisfied with their body image, with males having a higher dissatisfaction rate compared to females, who were more content with their bodies. Furthermore, the study revealed that 19.57% of the adolescents were underweight, 70.37% were of normal weight, and 9.87% were overweight/obese, showing significant variance in body-image perception [27].

Adolescence is a time when young people assert their independence, often rejecting their parents’ values [28]. The use of substances such as tobacco, alcohol, or other drugs is sometimes seen as an adult behavior or a form of rebellion. In this study, 29.3% of students reported starting smoking at the age of 13 or 14; 42% of girls had smoked at some point, compared to 19.6% of boys. Similar studies in comparable settings suggest that girls may use tobacco more frequently because it enhances their self-image and attractiveness [29,30]. A descriptive, cross-sectional study conducted in Jaén with a sample of 123 adolescents found that 42% had smoked at some point, despite 75% being aware of its harmful effects, particularly on respiratory health [9].

Alcohol consumption, according to this study, was significantly higher among girls than boys, with the most common settings for drinking being family parties, discos, or bars during weekends. While 31.5% of students reported having been drunk more than twice, no significant gender differences were found. Other studies on alcohol consumption in adolescents [8,31] have similarly concluded that alcohol use has increased in recent years, with consumption being more frequent on weekends, and higher among females than males in Spain.

In terms of drug use, a higher percentage of girls (23.2%) reported using hashish or marijuana compared to boys (13.7%). Girls also reported higher use of tranquilizers (11.6% vs. 5.9% for boys). Cocaine use was low and insignificant in both genders. In contrast, a study [26] observed a slight, non-significant decrease in cannabis use among girls, while use increased among 15–16-year-olds, possibly because they perceived it to enhance social interactions or improve mood [30].

These findings underscore the need for preventive programs focused on adolescent mental health, particularly within schools and families. Adolescents using substances reported a prevalence of sleep disorders of 29%, rising to 46% among smokers, while 31% of adolescents who consume alcohol reported insomnia [32,33]. A meta-analysis revealed that the overall prevalence of sleep disturbances in Chinese adolescents is 26% (95% CI: 24–27%), with a higher rate of 28% among senior high-school students, indicating an increase in sleep problems as students progress academically. Family- and school-based interventions have proven effective in not only reducing substance use but also in improving sleep quality and alleviating symptoms of depression and anxiety [34].

A notable finding is that alcohol consumption is significantly higher in family settings, such as parties (69.7%), compared to bars or nightclubs (51.7%). This suggests that alcohol use may be normalized for family celebrations, potentially lowering adolescents’ perceived risk associated with alcohol [35]. Kuntsche et al. [35] demonstrated that binge drinking among adolescents is strongly influenced by social contexts, particularly family environments where alcohol is often accepted. Similarly, Anderson and Baumberg [32] reported that family gatherings play a crucial role in shaping adolescents’ drinking patterns across Europe. These results underscore the importance of prevention programs that target both adolescents and their families, promoting an environment where alcohol is not central to family dynamics, especially for minors.

The results provide a detailed understanding of the mental health challenges and sleep disturbances among adolescents, complemented by a comparative analysis with prior research. This study concludes with evidence-based recommendations to strengthen mental health and substance use-prevention programs, with an emphasis on the role of schools and families in promoting overall well-being [35,36].

The findings reveal that 47.5% of adolescents report experiencing hopelessness about the future, while a similar percentage of them frequently feel bored. These indicators of psychological distress may reflect a disconnect from their environment and a lack of motivation, potentially linked to factors such as academic pressure or insufficient engagement in meaningful activities. Twenge and Campbell [37] observed a significant increase in depressive symptoms and hopelessness among adolescents after 2010, correlating these trends with rising digital media use and the superficial nature of online interactions. Similarly, Elhai et al. [38] noted that problematic smartphone use exacerbates feelings of boredom and hopelessness, two prevalent issues in adolescence [39,40].

### 4.1. Recommendations

Based on the findings of this study, several evidence-based recommendations are proposed to improve adolescent health outcomes, particularly in the domains of self-perceived health, emotional well-being, and substance use:

*Development of early preventive interventions*: Given the significant association between emotional distress and increased substance use, it is crucial to implement early preventive interventions targeting mental health and substance use in educational settings. Programs should emphasize emotional resilience, healthy body image, and coping strategies, while addressing the underlying factors contributing to substance use among adolescents. These interventions should be locally adapted and integrated into municipal health strategies to ensure that they reach the adolescent population effectively [34].

*Gender-specific approaches*: This study reveals marked gender differences in health perception, emotional regulation, and substance-use behaviors. Tailored interventions are needed to address these differences. Programs aimed at improving body image and managing emotional health should focus on the specific vulnerabilities of girls, while those aimed at boys should address patterns of substance use and emotional expression. Gender-sensitive approaches will ensure more effective and targeted outcomes [36]. Local health policies should also promote gender-specific programs within schools and communities.

*Integration of mental health services in schools*: The high prevalence of emotional distress, including fatigue, sleep disturbances, and depressive symptoms, indicates an urgent need for integrated mental health services within schools. Municipal authorities, in collaboration with educational institutions, should prioritize establishing school-based mental health support, such as regular screenings, access to counselors, and mental health education. These initiatives could mitigate the early onset of psychological distress and reduce the risk of substance use [34].

*Family and community engagement:* The role of the family and community in adolescent health is well-documented. Interventions should extend beyond schools to actively involve families and community stakeholders in promoting healthy behaviors and mental health awareness. Family-based interventions and community outreach, supported by municipal policies, can enhance the effectiveness of prevention programs by creating a supportive network for adolescents. Municipalities should foster collaboration between schools, health services, and community organizations to strengthen these initiatives [34].

*Policy recommendations for public health:* Policymakers, particularly at the municipal level, should prioritize the development of adolescent-focused public health policies that create supportive environments conducive to mental and physical well-being. These policies should include the regulation of substance availability, the creation of safe recreational spaces, and the expansion of accessible mental health resources. Furthermore, local governments should address socioeconomic disparities that influence adolescent health outcomes, ensuring that vulnerable populations have equitable access to health services and preventive resources.

*Addressing socioeconomic inequities:* The study highlights significant health disparities linked to socioeconomic status, with adolescents from lower socioeconomic backgrounds reporting poorer health and higher substance use. Public health initiatives, especially at the municipal level, must prioritize these populations by ensuring that preventive programs and mental health services are accessible and adapted to their specific needs. By targeting at-risk groups, municipalities can help reduce the health inequities observed in adolescent populations and promote a more inclusive approach to adolescent health.

*Comprehensive educational campaigns:* The widespread prevalence of substance use, particularly alcohol and tobacco, underscores the need for comprehensive, evidence-based educational campaigns. These campaigns should be multifaceted, targeting not only adolescents but also families and communities. Local governments can play a key role in promoting these campaigns, emphasizing the long-term health risks of substance use and promoting healthier alternatives for managing stress and emotional challenges.

*Longitudinal research on adolescent health:* Given the cross-sectional nature of this study, future research should adopt a longitudinal design to track changes in self-perception, emotional health, and substance use over time. Long-term studies will provide a deeper understanding of the developmental trajectories of these factors and offer insights into the long-term effectiveness of early interventions. Municipalities should collaborate with research institutions to gather local data and ensure that policies are continually updated based on evidence.

### 4.2. Study Limitations

This study analyzes how self-perceived health, mood, body image, and substance use are interrelated among 4th ESO students in both school and community contexts. By examining these factors alongside the influence of social and municipal policies, the study aims to provide a comprehensive understanding of adolescent health and well-being. Additionally, it emphasizes the importance of preventive programs that engage families and communities in creating environments that foster resilience and reduce risky behaviors.

However, certain limitations must be acknowledged. The lack of representativeness in the sample limits the generalizability of the findings to the broader adolescent population. Participants were drawn from a single school in the Baix Llobregat region of Catalonia, thus reducing the diversity in terms of socioeconomic, cultural, and geographic backgrounds. Future studies should aim to include more diverse and representative samples to ensure that the findings can be applied more broadly.

Moreover, the cross-sectional design provides only a snapshot of behaviors and attitudes at a single point in time, which limits the ability to assess long-term effects. Longitudinal studies would be valuable to track how these factors evolve and influence health outcomes over time.

Future research should also delve deeper into broader social and environmental determinants in different municipalities, such as family dynamics, socioeconomic resources, and *municipal public policy frameworks*. Understanding how local government policies impact adolescent health will be crucial for designing more effective, community-driven interventions that address the unique challenges faced by adolescents in different socio-geographic contexts. Addressing these systemic factors is essential for developing sustainable public health interventions that meet the specific needs of adolescents. Expanding the scope of research in these areas will further strengthen mental health and substance use-prevention strategies.

## 5. Conclusions

This study highlights the importance of early intervention in adolescent mental health and substance use. Adolescents with negative self-perception or emotional distress, such as anxiety and depression, are more prone to substance use, emphasizing the need for targeted prevention programs.

Gender differences were evident, with females being more vulnerable to anxiety and body image dissatisfaction, leading to higher substance use, while males showed greater alcohol consumption. Addressing body-image concerns, especially among females, could help prevent mental health disorders during adolescence.

The school environment and socioeconomic factors play a critical role in shaping adolescent health. Schools should be central to promoting mental health and reducing substance use, supported by community-based programs that enhance access to resources. Public health policies must address socioeconomic disparities, prioritizing vulnerable adolescents.

In conclusion, a comprehensive approach involving families, schools, and communities is essential to fostering resilience and prevent risky behaviors in adolescents. Public health strategies must consider emotional, familial, and community factors to effectively support adolescent well-being.

## Figures and Tables

**Table 1 healthcare-12-02304-t001:** Percentage distribution of self-perceived health by gender.

Category	Boys	Girls	*p*-Value
Very good	47.1% (n = 24)	39.1% (n = 27)	0.469
Good	43.1% (n = 22)	42.0% (n = 29)	0.879
Regular	7.8% (n = 4)	15.9% (n = 11)	0.095
Bad or very bad	2.0% (n = 1)	2.9% (n = 2)	0.723

**Table 2 healthcare-12-02304-t002:** Frequencies and percentages by emotional state.

Category	Always (n, %)	Often (n, %)	Sometimes (n, %)	Rarely (n, %)	Never (n, %)
Very tired to do things	18 (15.0%)	46 (38.3%)	38 (31.7%)	11 (9.2%)	7 (5.8%)
Having trouble sleeping	15 (12.5%)	23 (19.2%)	30 (25.0%)	29 (24.2%)	23 (19.2%)
Feeling displaced, sad, or depressed	7 (5.8%)	18 (15.0%)	40 (33.3%)	36 (30.0%)	18 (15.0%)
Hopeless about the future	13 (10.8%)	19 (15.8%)	25 (20.8%)	32 (26.7%)	32 (26.7%)
Nervous or tense	14 (11.7%)	35 (29.2%)	35 (29.2%)	22 (18.3%)	12 (10.0%)
Bored with things	21 (17.5%)	36 (30.0%)	38 (31.7%)	11 (9.2%)	12 (10.0%)
Angry or violent	0 (0.0%)	19 (15.8%)	43 (35.8%)	35 (29.2%)	18 (15.0%)

**Table 3 healthcare-12-02304-t003:** Gender distribution and statistical analysis of responses to questions about self-harm and suicidal thoughts.

Question	Response	Boys	Girls	*p*-Value
Have you harmed yourself intentionally?	Never	33 (70.6%)	20 (47.8%)	0.410
	Occasionally	14 (17.6%)	27 (46.4%)	0.496
	Frequently	1 (0.0%)	1 (1.4%)	1.000
	Almost every day	2 (2.0%)	0 (0.0%)	0.157
	Don’t know	3 (3.9%)	1 (1.4%)	0.479
	Prefer not to say	3 (5.9%)	2 (2.9%)	0.527
Have you thought about wanting to die?	Never	29 (58.8%)	16 (37.7%)	0.321
	Occasionally	15 (23.5%)	30 (39.1%)	0.454
	Frequently	4 (3.9%)	4 (5.8%)	1.000
	Almost every day	3 (5.9%)	1 (2.9%)	0.479
	Don’t know	0 (0.0%)	8 (5.8%)	0.005
	Prefer not to say	6 (7.8%)	5 (8.7%)	0.740

**Table 4 healthcare-12-02304-t004:** Analysis of body satisfaction, weight opinion, and BMI by gender.

Question/Response	Boys	Girls	*p*-Value
**Body satisfaction**			
Very satisfied	33 (70.6%)	20 (47.8%)	<0.001
Relatively satisfied	16 (33.3%)	46 (66.7%)	<0.001
Neither satisfied nor dissatisfied	8 (17.6%)	11 (27.5%)	<0.001
Relatively dissatisfied	3 (5.9%)	41 (59.4%)	<0.001
Very dissatisfied	1 (2.0%)	21 (30.4%)	<0.001
**Weight opinion**			
Too low	3 (5.9%)	3 (4.8%)	0.036
Slightly low	8 (15.7%)	9 (14.5%)	0.038
Appropriate	34 (66.7%)	36 (56.5%)	0.005
Slightly high	6 (11.8%)	9 (14.5%)	0.041
Too high	0 (0.0%)	5 (8.7%)	0.018
**BMI (Body Mass Index)**			
Underweight	6 (11.8%)	9 (14.5%)	0.042
Normal weight	39 (68.6%)	38 (60.9%)	0.025
Overweight	10 (17.6%)	9 (14.5%)	0.029
Obesity	0 (0.0%)	4 (7.2%)	0.032

**Table 5 healthcare-12-02304-t005:** Prevalence of smoking initiation and current smoking status by gender.

	Boys	Girls	*p*-Value
**Have you ever smoked?**			
Yes	24 (19.83%)	51 (42.15%)	<0.001
No	97 (80.17%)	70 (58.85%)	<0.001
**If you have smoked before, are you currently smoking?**			
Yes	10 (41.67%)	19 (37.25%)	0.539
No	14 (58.33%)	32 (62.75%)	0.539

**Table 6 healthcare-12-02304-t006:** Prevalence of alcohol consumption by gender.

Have You Ever Drunk Alcohol?	Boys	Girls	*p*-Value
Yes	56.9% (34)	82.6% (50)	0.003
No	43.1% (26)	15.9% (11)	0.003
**In Which Occasions Do You Drink?**	**Yes**	**No**	** *p* ** **-Value**
During class days	0%	37%	<0.001
During weekends	21%	27%	0.001
At family parties	59%	11%	<0.001
At discos/bars	43%	17%	0.001
**Have You Been Drunk?**	**Boys**	**Girls**	** *p* ** **-Value**
Been drunk and 4 or more times	34.5% (25)	35.1% (17)	0.939
Been drunk	24.1% (17)	21.1% (10)	0.637
4 or more drinks	0% (0)	5.3% (3)	0.216
Never drunk	31% (22)	33.3% (16)	0.751
NS/NC (no response)	10.3% (7)	5.3% (3)	0.271

**Table 7 healthcare-12-02304-t007:** Prevalence of use of various addictive substances by gender.

Substance Use	Boys	Girls	*p*-Value
Hashish or marijuana	13.7% (10)	23.2% (11)	0.029
Tranquilizers	5.9% (4)	11.6% (6)	0.083
Cocaine	0% (0)	1.4% (1)	0.222
Speed/amphetamines	0% (0)	0% (0)	N/A
Ecstasy	0% (0)	1.4% (1)	0.222
Inhalants	0% (0)	2.9% (2)	0.135
No substance	78.4% (57)	66.7% (32)	0.073

## Data Availability

The data presented in this study are available upon request from the corresponding author.

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
