# Peer review of "Self-Perceived Health, Mood, and Substance Use Among Adolescents: An Analysis to Enhance Family, Community, and Mental Health Care"

_healthcare, 2024, doi:10.3390/healthcare12222304_

Round 1

Reviewer 1 Report

Comments and Suggestions for Authors

Pleas mentioned the time and place in the title.

In the abstract, it is better to make the introduction shorter.

... : A descriptive crosssectional, but you have statistical analysis , it’s mean analytical study…

In the abstract, the sampling method should be stated.

In the abstract, a brief description of the sample and their situation should be presented.

In the abstract, the conclusion should be based on the results.

If possible, the introduction should be more concise.

In the introduction, all sentences should be referenced.

Is the sample size sufficient?

What is the basis for calculating the sample size?

Explain the sampling method more clearly.

The sample size is not enough.

The contents of the tables do not need to be completely rewritten in the text.

For table 1, one p value  is not enough(chi square)?

The results are too long, boring and without summarization.

Tables are confusing.

The discussion is not based on hypotheses, it is long.

The conclusion is long.

Author Response

I greatly appreciate your valuable comments and observations regarding our manuscript. Below, we address each of the points you mentioned: 

Abstract: We have adjusted the introduction of the abstract to make it more concise and direct. Additionally, we have added a brief description of the sampling method and the sample, as per your recommendation. The conclusion now focuses more directly on the results obtained. 

Sampling method: We have clarified the sampling method used, providing more details about its basis and justification, and explaining the criteria for the sample size. While we understand your concern about the sample size, we believe it is sufficient for the study's objectives, as it includes the entire population of this school in a municipality of 13,000 in habitants (there is only one school), and it is coordinated with the public health department of the town hall. However, we are open to further discussion if you deem it necessary. 

Statistical analysis: Regarding your observation about the statistical analysis, we have revised the text to specify more accurately that this is an analytical study with mean analyses, in addition to descriptive analyses. 

Tables: We appreciate your observation about the tables. We have reviewed the tables to reduce complexity and avoid redundancy in the text. We have also reconsidered the use of p-values for the relevant comparisons. 

Results and discussion: We have shortened and synthesized the results to make them clearer and more concise. The discussion has been restructured to better align with the hypotheses and reduce its length. The conclusion has also been revised to be more direct and precise, focusing exclusively on the main results. 

We are very grateful for your time and suggestions, which have undoubtedly improved the quality of our work. We remain at your disposal for any further adjustments you may consider necessary

Reviewer 2 Report

Comments and Suggestions for Authors

Author Response

Dear reviewer, 

I sincerely appreciate your positive evaluation of the article and your constructive comments. Your observations are very valuable and will contribute to improving the quality of our work. Below, I address each of the points you mentioned: 

  1. Importance and Timeliness of the Study: Thank you for your comment regarding the need to establish why this study is timely. We will incorporate a clearer discussion on the current relevance of the study in the context of the challenges faced by adolescents today. 
  1. Terminology: We understand your preference for the term "dependence" instead of "addiction." We will make the necessary changes in the manuscript to reflect this more neutral terminology. 
  1. Analysis of a Single Year Group: We will justify the choice to analyze a single year group, indicating that this was done to maintain a specific and in-depth focus on a homogeneous population in a given context. It should be noted that this is the first time that the study has been carried out in this population. 
  1. Sample Size: We appreciate your observation regarding the sample size. We will explain that the choice of 121 participants is based on the fact that it represents the total population of the only school in a municipality of 13,000 inhabitants. While we acknowledge that it may seem limited, we believe it is representative of the study's context. 
  1. Formatting Issues: We will review the presentation of the article to address the font issues after Table 1 and ensure consistency in the presentation of percentages. We will also correct the division of sentences in paragraphs on pages 5 and 7. 
  1. Coherence in Language: Thank you for your comment regarding coherence in language use. We will make the necessary adjustments to ensure that the vocabulary used is consistent throughout the document, especially in the introduction and discussion. 
  1. Global Contextualization: We find your suggestion to situate our findings in a global context relevant. We will incorporate a discussion that links our results to international research on these topics. 
  1. Exploration of Interesting Findings: We appreciate your observation about the relationship between substance use and body image, as well as the impact of the COVID-19 pandemic. We will explore these topics in greater depth in the discussion, evaluating how they may have influenced the participants' perceptions. 
  1. Material on Socioeconomic Status: We will add information on how socioeconomic status is addressed in the study and how it relates to our findings. 
  1. Opportunity to Explore Findings: We appreciate your observation about the need to delve deeper into our findings. We will ensure that our recommendations and conclusions are supported by a more detailed analysis of the results. 

Once again, we thank you for your time and suggestions, which will undoubtedly enrich our manuscript. We remain at your disposal for any further adjustments you may consider necessary. 

Sincerely 

Reviewer 3 Report

Comments and Suggestions for Authors

First of all, I would like to congratulate you for the subject of your study and the interest of the results obtained. I would like to highlight the quality of the introduction, which supports the object of study of the research with relevant documentation. However, your article presents some important weaknesses in terms of methodology, which I will now detail. Firstly, the lack of representativeness of the sample should be taken into account, considering that it is a cross-sectional study. This issue would not arise if it were a longitudinal study

Additionally, the instrument used is not described, nor is it indicated whether it is a previously validated questionnaire, or one that has been validated and used in other scientific research. This makes it impossible for other researchers to replicate the study. 

The questionnaire administered does not indicate that any method has been used to identify and exclude from the analysis those questionnaires that were completed randomly, dishonestly, or pseudo-randomly, such as the Oviedo Scale of Response Infrequency, formed by elementary and dichotomous (yes or no) questions. Fonseca E, Paíno M, Lemos S, Villazón Ú, Muñiz J. Validation of the Schizotypal Personality Questionnaire Brief form in adolescents. Schizophr Res. 2009;111:53-60.

I encourage you to continue working in this direction and to address the suggestions provided.

Sincerely

Author Response

Dear Reviewer, 

I sincerely appreciate your positive comments regarding the topic of our study and the quality of the introduction. Your observations are very valuable and will contribute to improving our manuscript. Below, I address the points you mentioned: 

  1. Representativeness of the Sample: We understand your concern regarding the lack of representativeness of the sample in a cross-sectional study. In the manuscript, we will emphasize the specificity of the context and justify our approach by explaining that the sample corresponds to the entirety of the only institute in a municipality with 13,000 inhabitants, which provides significant representation in that context. Highlighting that the total number of ESO students is 563. 
  1. Instrument Used: We regret not having provided sufficient information about the instrument used. In the revision, we will include details about the questionnaire as well as the bibliographic citation and confirm that it is a validated instrument that has been utilized in previous research. This will allow other researchers to replicate the study. 
  1. Identification and Exclusion of Invalid Questionnaires: We appreciate your observation regarding the need to indicate the methods used to identify and exclude questionnaires completed randomly or dishonestly. In our study, we utilized a questionnaire provided by the Diputació de Barcelona, which has conducted surveys from 2015 to June 2024 in 99 municipalities within the region, with a participation of 34,960 students. Given that we are evaluating a single municipality, the commission focuses on conducting promotion and prevention activities specifically for that community. Therefore, we selected questions that align with our research objectives. In the revised version, we will detail the procedures implemented to ensure the quality of the collected data, including the methods applied to identify and exclude invalid questionnaires. 
  1. Appreciation for Suggestions: We thank you for your encouragement to continue working in these directions, and we will take your suggestions into account to strengthen the study's methodology. 

Once again, I thank you for your time and valuable comments, which will undoubtedly enhance the quality of our work. We remain at your disposal to make further adjustments if deemed necessary. 

Sincerely, 

Round 2

Reviewer 2 Report

Comments and Suggestions for Authors

Thank you for your thoughtful changes.

You have changed pieces that I didn’t ask for but frustratingly I can’t find the other reviewer’s comments to see what you were responding to.

You have dealt with the formatting and language issues as requested.

Your new paragraph in the Discussion has a typo and you need to reference the material.

Your reworking of the recommendations was good.

This paper continues to have a Spanish-centric focus.

I think there is always a danger as a reviewer to ask for too much.  The Discussion is ok, & I would have put the emphasis elsewhere, but I realise that you are restricted by the word count.

Change the COVID paragraph & I am happy for this to be published.

Author Response

Thank you for your comments and for your appreciation of the changes made. I regret that you were unable to find the comments from other reviewers; I have attempted to address all the points raised. We have reviewed the new paragraph in the Discussion and, considering it irrelevant, we have removed it. I appreciate your observations on the recommendations and the discussion; I have tried to adapt to the word limit set. I am pleased that the article is now ready for publication.

Sincerely, Dr. Jose Antonio Zafra Agea

Reviewer 3 Report

Comments and Suggestions for Authors

The authors' contributions after the comments made in the review respond to what was requested. Just one small shorthand detail to correct: in the abstract there are two full stops on the second line.

Kind regards

Author Response

Thank you for your comments. I am glad that the contributions made have addressed the requests. Regarding the detail noted in the abstract, the two colons in the second line have been corrected.

Sincerely, Dr José Antonio Zafra